# Microneedles: An Emerging Vaccine Delivery Tool and a Prospective Solution to the Challenges of SARS-CoV-2 Mass Vaccination

**DOI:** 10.3390/pharmaceutics15051349

**Published:** 2023-04-27

**Authors:** Ya-Xiu Feng, Huan Hu, Yu-Yuen Wong, Xi Yao, Ming-Liang He

**Affiliations:** 1Department of Biomedical Sciences, Jockey Club College of Veterinary Medicine and Life Sciences, City University of Hong Kong, Hong Kong SAR, China; yxfeng5-c@my.cityu.edu.hk (Y.-X.F.); huanhu4-c@my.cityu.edu.hk (H.H.); yuywong5-c@my.cityu.edu.hk (Y.-Y.W.); 2CityU Shenzhen Research Institute, Shenzhen 518071, China

**Keywords:** immune response, microneedles, mass vaccination, SARS-CoV-2 vaccine, vaccine delivery

## Abstract

Vaccination is an effective measure to prevent infectious diseases. Protective immunity is induced when the immune system is exposed to a vaccine formulation with appropriate immunogenicity. However, traditional injection vaccination is always accompanied by fear and severe pain. As an emerging vaccine delivery tool, microneedles overcome the problems associated with routine needle vaccination, which can effectively deliver vaccines rich in antigen-presenting cells (APCs) to the epidermis and dermis painlessly, inducing a strong immune response. In addition, microneedles have the advantages of avoiding cold chain storage and have the flexibility of self-operation, which can solve the logistics and delivery obstacles of vaccines, covering the vaccination of the special population more easily and conveniently. Examples include people in rural areas with restricted vaccine storage facilities and medical professionals, elderly and disabled people with limited mobility, infants and young children afraid of pain. Currently, in the late stage of fighting against COVID-19, the main task is to increase the coverage of vaccines, especially for special populations. To address this challenge, microneedle-based vaccines have great potential to increase global vaccination rates and save many lives. This review describes the current progress of microneedles as a vaccine delivery system and its prospects in achieving mass vaccination against SARS-CoV-2.

## 1. Introduction

The invention of vaccines and vaccination is one of the greatest public health achievements. Vaccines can mimic infection and elicit specific acquired immunity against disease infection [1]. The World Health Organization (WHO) estimates that vaccination can prevent 3.5 to 5 million deaths each year [2]. As a result, vaccination is one of the most powerful tools for protecting people from infectious diseases. Despite the success of the vaccine era, currently, around 15 million people worldwide die each year from various infectious diseases [3]. The major bottleneck is the strict low-temperature conditions required for the storage and transportation of vaccines. Another barrier is the shortage of professional medical personnel in some areas [4]. Portnoy et al. calculated that the cost of vaccination in 94 low- and middle-income countries over a ten-year period from 2011 to 2020 was about USD 62 billion, of which the delivery cost and supply chain cost were USD 34 billion and USD 4 billion, accounting for 54% and 6% of the total cost, respectively [5]. The high cost of vaccines has seriously hindered vaccine coverage in these countries. Their lack of professional medical staff further restricts the large-scale vaccination. Currently, syringe injection into the subcutaneous tissue or muscle is the most common way for vaccination. In fact, due to pain, needle shaft damage, needle reuse, and poor patient compliance, a large proportion of people reject vaccination, which limits vaccine coverage [6]. Therefore, a novel development of vaccines and mass vaccination remain top priorities for WHO.

To overcome the problems mentioned above, microneedle (MN) delivery of vaccines has come into the spotlight. MNs consist of a series of needles with a length controlled within 1 mm and a bottom layer, which can easily pierce the stratum corneum by thumb pressing or by an applicator to deliver the vaccine into the skin (Figure 1). The needle tips of MNs do not penetrate into the subcutaneous tissue and do not touch blood vessels and nerves, which overcomes the problems of pain and wound bleeding caused by traditional syringe injection [7]. Meanwhile, the skin layers beneath the stratum corneum are supported by a tight junction of immune-network-supporting response APCs, and the number of APCs in the epidermis and dermis is much greater than that in muscles, including Langerhans cells (LCs) and dendrite cells (DCs). LCs represent 1–5% of all epidermal cells, which are located mainly at the suprabasal layer of the epidermis [8]. Meanwhile, DCs constitute a major part of the dense APC network in the dermal layer. Muscle contains only a small number of APCs, and humoral immunity mainly occurs with the intramuscular vaccine [9]. The time difference between humoral immunity and cellular immune response provides opportunities for virus transmission and infection [10]. Skin vaccination has proven to induce similar antibody titers using doses lower than those used for the intramuscular (i.m.) route (that is, a dose-sparing effect) [11]. Liu et al. proved that antigen ovalbumin (OVA)-loaded NPs delivered via a hollow microneedle array (hMN) elicited a significantly higher IgG2a antibody response in comparison to antigen-loaded NPs delivered by intramuscular injection [12]. Although a vaccine using the Mantoux technique also uses an intradermal route, it has a higher frequency of pain and itching than the intramuscular group [13]. The Bacillus Calmette-Guérin (BCG) vaccine is the only Essential Programme on Immunization (EPI) vaccine that is given intradermally. Hiraishi et al. reported the design and engineering of a BCG-coated microneedle vaccine patch for a simple and improved intradermal delivery of the vaccine. The response was comparable to the traditional intradermal BCG vaccination, which could facilitate increased coverage, especially in developing countries that lack an adequate healthcare infrastructure [14]. In addition, with the exception of the hMN, most MN-based vaccines are in an anhydrous form and reduce reliance on cold chain by limiting the vibration of molecules to heat, allowing stable storage of vaccines [15,16].

MN-based vaccines can also be conducted by self-injection, which greatly improves the vaccination rate in special populations. Table 1 compares the differences between MN-based vaccines and traditional injection vaccines.

Several important milestones in MN development are illustrated (Figure 2). The first study of sMN immunization was conducted in 2002. Mikszta et al. used a novel micro-enhanced array (MEA) to break the skin barrier and deliver the vaccine locally in vivo for the first time [25]. MEA-based topical vaccines are more inducible than needle injections and require fewer vaccinations. This method effectively allows the vaccines to penetrate directly into the dermis, reducing skin irritation and eliminating pain and discomfort. In 2010, the first dMN were successfully used for influenza vaccination in the mice skin, which was well tolerated and generated a strong antibody response [26]. Then, in 2015, an dMN-based vaccine was first tested in non-human primates [27]. With the deepening research, the application of MN vaccines has become more and more extensive and includes the influenza vaccine, rabies vaccine [28], HPV vaccine [29] and so on. Now, researchers are applying MNs to the delivery of the COVID-19 vaccine [30].

The coronavirus disease 2019 (COVID-19) pandemic has so far resulted in 625 million confirmed infections and more than 6.5 million deaths worldwide, which has overwhelmed health and economic systems [31]. While candidate vaccines against Severe Acute Respiratory Syndrome Coronavirus 2 (SARS-CoV-2) are rapidly being developed, achieving herd immunity remains a major obstacle for the immunization campaigns [32]. At present, problems such as incomplete vaccine manufacturing capacity, high cost and strong dependence on cold chain supply have seriously hindered the popularization of vaccines [33]. MN delivery of COVID-19 vaccines has the potential to address these issues, and it is likely to be an alternative to traditional injection vaccines, especially for special populations, which may greatly improve vaccine coverage.

This review systematically summarizes the development process of MNs, which overcome the limitations of traditional vaccination and provide a new way for vaccine delivery. We detail the possibility and utility of MNs for administering different types of vaccines, and we reveal their great potential to address the challenges of mass vaccination against COVID-19, especially in special populations.

## 2. Development and Selection of MNs

MNs are a new type of physical penetration-enhancing technology that consist of multiple micron-level fine needle tips connected to the base in an array, or they may entail just one or several micro-needles connected to standard syringes, vials and carpules [34,35]. The needle body is generally 10–2000 μm in height and 10–500 μm in width. The advent of MN vaccination aims to make up for the inadequacies of traditional syringe injections. MNs penetrate the stratum corneum of the skin in a minimally invasive way to form reversible microtubule channels, allowing substances to reach the epidermis or dermis and spread through blood vessels, finally achieving systemic transmission [36]. MNs are used in the administration of drugs and vaccines, in the cosmetic field, and in the treatment and diagnosis of diseases.

In 1958, Wagner first proposed the concept of MN intradermal injection and applied it for patent protection [37]. In 1976, Gerstel and Place further proposed another patent for using MNs for transdermal drug delivery. They conceived a device that includes a drug-containing reservoir and multiple piercing protrusions extending from the surface of the reservoir that can pierce the stratum corneum of the skin to enhance transdermal drug delivery [38]. However, the MN device had not received any practical application. Until 1995, an application breakthrough was achieved to prepare MN arrays on silicon wafers for the first time. Hashmi et al. used a microinjection to introduce foreign genes to Heterorhabditis bacteriophora HP88 [39]. In 1998, Henry and others applied MNs to transdermal drug delivery and eventually achieved industrialization [40]. Then, in 2002 [25], Mikszta et al. opened a new chapter in the MN delivery of vaccines. The first human study with vaccine MNs appeared in 2009 [41], and the first dMN-based vaccine was obtained in the following year [42]. In 2014 and in the clinical trial of the MN vaccine in following year, Ma et al. delivered an HIV vaccine into the oral cavity using cMN to induce systemic and mucosal immune responses. Lastly, the FDA published a Regulatory Consideration for micro-needling devices in 2017 (Figure 2).

### 2.1. Classification of MNs

MNs generally fall into five different depending on their delivery strategies: solid, coated, hollow, dissolving, and hydrogel-forming (Figure 3).

Solid MNs were the first generation and were made of metal materials and non-degradable polymers such as silicon and titanium dioxide. They are generally described as “poke and patch” due to their inability to carry the drug [43]. They can pierce the skin, leaving a microchannel that allows drugs or vaccines to reach the dermis. Coated MNs were proposed to overcome the complexity of the two-step process in solid MNs [44]. They load the active substances through the coating on the surface of the needle body, then release the deliverables into the capillaries through the intercellular fluid, and finally induce systemic treatment. They are described as “Coat and poke”, and the substances delivered are mainly water-soluble, which can be reused [45]. An example is the successful preparation of smallpox vaccine-coated MNs by Chou et al., which turned out to be an alternative delivery system for traditional smallpox vaccination and storage [46]. The hollow MNs can be understood as a microsyringe due to the cavity in the middle of the needles. When they pierce the skin, the vaccines preloaded in the cavity are driven by the concentration gradient to achieve delivery. They are usually made of hard materials such as silicon, metals and polymers [47].

All three types of MNs mentioned above are at risk of breakage and generate sharps waste, which prompted the development of dissolving MNs [48]. They are made of water-soluble biocompatible/biodegradable polymers or sugars. Dissolving MNs are often described as “Poke and release”. After penetrating the skin, the needles gradually dissolve in the skin tissue, and the substances contained in the needles are gradually released and absorbed by the body. Because of their advantages in delivering various therapeutic agents and vaccines, dissolving MNs have become widely used throughout the MN industry. Hydrogel MNs are usually made of cross-linked hydrogels or super-swollen polymers [49]. They are used to deliver drugs by swelling or to diagnose by absorbing certain substances from interstitial fluid in the skin without dissolving.

### 2.2. Materials for the Preparation of MNs

The main attribute of MNs is their ability to penetrate the skin without breaking or bending. At present, the materials used for MNs are divided into three categories: inorganic materials, metals, and polymers (Figure 3).

Inorganic materials used in MNs include mainly silicon, glass, and ceramics. Silicon is the first and most frequently used material, which can be customized in different shapes and sizes for a wide range of applications. Although it has high hardness and can be easily inserted into the skin, silicon is brittle and can cause adverse effects if left in the skin [50]. Because of this, scientists have used glass as a material to make MNs. Glass is now widely used for alumina, zirconia and calcium sulfate hemihydrate [51,52,53]. However, glass is brittle and fragile. Scientists have begun to investigate biocompatible ceramic materials. For example, Vallhov et al. evaluated biodegradable ceramic (calcium sulfate) MNs and found that they could release drugs without triggering an immune response [54].

Metals have been used in medical devices for decades. Biomedical metallic materials have good biocompatibility; gold, silver, platinum, stainless steel, cobalt-based and titanium-based alloys, etc. have mechanical properties useful in dentistry. Nickel-chromium stainless steel, cobalt-chromium-molybdenum alloys, titanium and their alloys are used in orthopedics, which are suitable materials for MNs. Among these, stainless steel, titanium, and nickel are often used for metal MNs [55,56,57]. However, due to high hardness, non-biodegradability and inorganic materials, metals can only be used for solid, coated, or hollow MNs. Scientists further developed high molecular polymers to make dissolving and hydrogel MNs.

Polymer is a very promising material for MNs. Various biocompatible and biodegradable polymers have also been used to prepare MNs, such as hydrophobic poly (methyl methacrylate) (PMMA), Polyvinyl alcohol (PVA), Polyvinylpyrrolidone (PVP), polycaprolactone (PCL) and hydrophilic hyaluronic acid (HA), Carboxymethyl Cellulose (CMC), Polyethylene glycol (PEG) et al. [58]. Polymers are not as rigid as silicon and metals, and they are commonly used to make dissolving and hydrogel MNs [59].

## 3. Development of MNs Delivery of Vaccines

The development of society and technology is driving the progress of vaccines. In addition to the basic requirements of vaccines, such as sterility, safety and efficacy, modern vaccines also need to have the characteristics of low cost, high heat stability, convenient injection, and long-term immunity. MNs delivery of vaccines may meet those needs.

### 3.1. Classification of Vaccines

Vaccines approved for human use in the pharmaceutical industry today fall into four main subcategories: live-attenuated vaccines, inactivated vaccines, pathogen component vaccines (DNA, RNA, proteins, virus-like particles (VLPs), etc.), and toxoid vaccines. Among them, live attenuated and inactivated vaccines are whole-agent vaccines, and pathogen component vaccines are subunit vaccines (Figure 4).

Currently commercialized vaccines are designed and produced based on these four types, as shown in Table 2.

### 3.2. MNs Deliver Different Types of Vaccines

Mikszta et al. [25] first tried MN immunization with a novel MEA in 2002 and successfully broke the skin barrier to deliver the vaccine to the body. Research on the delivery of vaccines via MNs has been intensively pursued after that (Figure 5).

#### 3.2.1. Live-Attenuated Vaccine

Live-attenuated vaccines are whole vaccines in the form of weakened “wild” viruses or bacteria. When introduced into the body, they can trigger an immune response as with normal bacteria or viruses without causing harm or disease. However, these weakened forms are very likely to mutate back to the former wild type and cause fatal damage to the organism. Oral poliovirus vaccines, for example, can revert to their original toxic form, which can enter the central nervous system and cause paralysis in patients [60]. In addition, vaccines are highly unstable at temperatures above 8 °C. Professional operations are also required to maintain the cold chain for vaccine storage and transport. The use of MNs can reduce the need for low temperatures while ensuring a more stable state of vaccines, and it is safer and more convenient to operate.

In a study using the attenuated Japanese encephalitis vaccine ChimeriVax in non-human primates, it was found that the same dose of vaccine administered via skin micro abrasion and penetration with MNs produced stronger immunity than from subcutaneous injection [73]. In addition, Edens et al. used dissolving MNs to deliver a live-attenuated measles vaccine to rhesus monkeys and found that MN vaccination produced higher antibody titers than subcutaneous vaccination. In addition, compared with commercially available freeze-dried and liquid vaccines, which can be stored for 90 days and only 7 days, respectively, the MN measles vaccine can retain about 90% of its potency after being stored at 40 °C for 4 months [27]. Prausnitz et al. also summarized the benefits of using MNs for measles and rubella vaccination, including the simplification of supply and cold chain, the elimination of needlestick waste, and the reduction of vaccination system costs. Hiraishi et al. used a microneedle patch (MNP) to efficiently deliver a BCG vaccine into the epidermis and dermis of the skin and elicit a robust cell-mediated immune response in the lungs and spleen of guinea pigs. This approach not only simplifies logistics and eliminates the hazards posed by hypodermic needles, but also promises to increase BCG vaccination rates [14].

#### 3.2.2. Inactive Vaccine

Inactivated vaccines are produced by inactivating intact viruses or bacteria by chemical methods (beta-propiolactone, formalin, etc.) or physical methods (ultraviolet rays, electron beam irradiation, etc.) [74]. Although this kind of vaccine is not effective as the live-attenuated vaccine, it is safer. Inactivated vaccines can induce massive immune responses by multiple injections or co-injection with adjuvants. Hepatitis A, hepatitis B, and influenza vaccines are all inactivated vaccines [61]. To date, there are many examples of the successful use of MNs to deliver inactivated vaccines.

Hirschberg et al. used dissolving MNs to inject a hepatitis B vaccine containing aluminum hydroxide and lipopolysaccharide (LPS) adjuvant, and its primary immunization effect was comparable to the secondary immunization effect of a conventional vaccine containing liquid alum adjuvant [63]. Moreover, the conventional vaccine lost 40% of its antigenicity after one week at 50 °C, whereas the MN vaccine did not significantly decrease after three weeks at 50 °C. This example successfully demonstrates that MN may be the most promising alternative to needle injection for hepatitis B vaccines. Frewab et al. evaluated the acceptability of an inactivated influenza vaccine delivered by MNP compared with an inactivated influenza vaccine (IIV) delivered by hypodermic needle [64]. They screened 112 normal people in Atlanta, Georgia, and they found that participants also preferred the MNP for influenza vaccination and follow-up immunizations. Rodgers et al. successfully inserted and dissolved MNs ranging in size from 254 to 381 microns into the skin of mice. The results demonstrated that the bacterial load in the lungs of mice that had been inoculated with Pseudomonas aeruginosa was significantly lower than that of uninoculated mice. This work suggests the potential of dissolving MNs for intradermal vaccination against heat-killed bacteria [65].

#### 3.2.3. Pathogen Component Vaccine

Pathogen vaccines are subunit vaccines and contain antigenic parts of viruses and bacteria. It can trigger an immune response without harming the subject [75]. Pathogen vaccines include DNA, RNA, protein, and VLPs, and they resemble inactivated vaccines but are less immunogenic than live viruses or bacteria. They are safer but require multiple doses to achieve the desired level of the immune response [76].

##### DNA Vaccine

DNA vaccines have been studied since the early 1990s. They are antigen-encoding plasmid vectors containing a gene of interest [77]. When the plasmid is transfected into myocytes or inoculated into APCs in skin or muscle, it triggers the transcription of the gene and the production of an antigenic protein, eliciting antigen-specific immune responses in vivo [78]. The advantages of DNA vaccines are that they are more heat-resistant than traditional forms of vaccines and can be easily mass-produced [79]. DNA vaccines cannot be reverted to their original virulent form [80].

Arya et al. evaluated the safety and immunogenicity of MNP vaccination with a rabies DNA vaccine [81]. The vaccine in the MNP was stable at 4 °C for at least three weeks, and the MNP was well tolerated in the skin, with complete resolution of skin reactions within seven days and no systemic side effects. The immunogenicity of the MNP outperformed the intramuscular injection of the same vaccine dose. Dissolving MNPs may provide an innovative approach for mass rabies vaccination [28]. Cole et al. used MNs to enhance the immunogenicity of DNA vaccines [82]. This approach not only improves DNA stability in solid matrices, but also increases DNA delivery ability compared to sMN. To achieve an effective transdermal vaccine and targeted delivery in developing countries, Hu et al. utilized the MN delivery of a DNA vaccine for the treatment of malignant melanoma. The results showed that the MN-delivered vaccine induced significant therapeutic anti-tumor immunity and inhibited cancer cells growth, which is a potential immunotherapy strategy for MM [66]. Liao et al. synthesized DNA multiplex vaccines in a single step in an MNP. It can be stored at 45 °C for at least 4 months, which has a significant impact on effective vaccination in developing countries [83]. In addition, Qiu et al. developed a hepatitis B DNA vaccine system based on a soluble microneedle array that can induce an effective immune response [67].

##### RNA Vaccine

RNA vaccines function by introducing an mRNA sequence that encodes a specific antigen [84,85]. The vaccine introduces molecules of antigen-encoding mRNA, and the designed mRNA serves as a blueprint for building a foreign protein that would normally be produced by a pathogen (e.g., a virus) or by a cancer cell. These protein molecules stimulate an adaptive immune response that teaches the body to identify and destroy the corresponding pathogen or cancer cells [60]. The mRNA is delivered by a co-formulation of the RNA encapsulated in lipid nanoparticles that protect the RNA strands and facilitate their uptake into cells [86].

Since Nair and Boczkwoski first successfully demonstrated mRNA-based cancer vaccination in 1996 [87], scientists have been searching for the best mRNA delivery system in recent years. MNs have attracted considerable interest as a platform for delivering vaccines through the skin. Koh et al. reported a proof-of-concept study to produce, characterize, and therapeutically evaluate in vitro transcribed messenger RNA (mRNA) loaded into dissolving MNP (RNA patches) [88]. They found that the physical and functional integrity of mRNA stored in the MNs were preserved for at least two weeks. The RNA MNP can mediate in vivo transgene expression of mRNA encoding luciferase for up to 72 h, and the transfection efficiency and kinetics are superior to subcutaneous injection. Golombek et al. evaluated the intradermal delivery of synthetic mRNA by injection with hollow MNs. In addition, an in vitro porcine skin model was established to analyze protein expression mediated by synthetic mRNA in the skin after intradermal administration [89]. Using this model, the efficient delivery of synthetic mRNA was demonstrated to detect high levels of secreted humanized Gaussian hypolucidase (hGLuc) protein encoded by microinjection of synthetic mRNA. The use of MNs enables the patient-friendly, painless, and efficient delivery of synthetic mRNA into the dermis. This approach can be used for topical treatment of different skin diseases as well as for vaccination and immunotherapy. There are only two typical examples of RNA vaccine delivery using MNs so far. However, in the recent fight against COVID-19, the use of MNs to deliver RNA vaccines has been widely used, and we will present these in detail in Section 4.

##### Protein Vaccine

Protein subunit vaccines contain specific isolated proteins from viral or bacterial pathogens to trigger protective immunity [90]. Rather than injecting a whole pathogen to trigger an immune response, a protein vaccine is safer and more stable but more complex to manufacture. Generally, adjuvants are required to induce a strong immune response, and multiple injections are performed.

MNs can be used as a delivery tool to deliver protein vaccines efficiently. Yuan et al. delivered the F1 protein antigen of Yersinia pestis in MNs, which were successful in triggering an immune response against the plague in animals [69]. Weldon et al. tested the hypothesis that a recombinant subunit influenza vaccine could be delivered to the skin via coated MNs [70]. It was found that mice vaccinated with stable recombinant trimeric soluble hemagglutinin (sHA) by MN elicited a strong immune response. The mice were completely protected from lethal influenza virus infection, highlighting the benefits of this protein subunit vaccination. Wang et al. generated self-adjuvant protein nanoparticles that conserved influenza antigens and immunized mice by vaccinating the skin with dissolvable MNPs to enhance the strength and breadth of the immune response [91]. They produced a bilayer protein nanoparticle, NA2-FliC/M2e, and demonstrated that this nanoparticle-based MNP skin vaccine could be developed as an independent or synergistic component of a universal influenza vaccination strategy.

##### VLP Vaccine

VLPs are nanoparticles composed of a subset of non-infectious viral components that are structurally similar to wild-type viruses but lack the viral genome. They are non-replicative and non-infectious, and can induce an immune response in the host. VLPs have good stability and are excellent vaccine carriers [92,93].

Quan et al. used coated MNs to deliver influenza VLPs into mouse skin [26]. It was found that the delivery of high doses of vaccine via MNs resulted in 100% protection against challenging influenza viruses. In contrast, unstable influenza VLPs and intramuscular vaccines weakened the immune system and provided only partial protection (≤40%). A vaccine formulated with a coated MNP was shown to provide superior protection over intramuscular injection through dermal vaccination with potential dose maintenance. Ray et al. developed dissolving MNs that contained a candidate HPV vaccine consisting of Qβ VLPs [23]. Compared with conventional subcutaneous injection, polymer MN delivery of Qβ-HPV produced similar levers of anti-HPV16 L2 IgG antibodies, with a lower (16.7%) intradermal dose required. In addition, the vaccine can remain stable in the MNs at room temperature for several months, which will effectively solve many problems related to the cold chain. This MNP vaccine not only saves vaccine doses, but is also easy to self-administer and minimally invasive, which enables the wide distribution of HPV vaccines and improves patient compliance. The Qβ-VLPs and their MN delivery technology are a plug-and-play system that can serve as a general platform with a wide range of applications. Guo et al. developed a novel tumor vaccine delivery strategy using a biodegradable microneedle patch (MN) that allows for the sustained release of tumor antigens and induces long-term antitumor responses [94]. Kines et al. also demonstrated that mice immunized with HPV16 VLP-coated microneedles produced a robust neutralizing antibody response, and the MN delivery of freeze-dried HPV may provide a practical, heatable vaccine delivery method that can be evaluated clinically [29].

#### 3.2.4. Toxoid Vaccines

Toxoid vaccines use toxins (harmful products) produced by disease-causing bacteria that develop immunity against the disease-causing part of the bacteria rather than the bacteria themselves. A toxoid is an inactive toxin that has lost its ability to cause disease but retains its immunogenicity [95]. Two vaccines contain toxoids as immunogens, namely the diphtheria and tetanus vaccines. Like some other vaccines, the toxoid vaccines require booster shots to provide lasting protection against disease. In recent years, numerous studies have been conducted on the use of microneedles to deliver toxoid vaccines [96].

Groot et al. tested ceramic nanoporous microneedle arrays (MNAs), which are a novel MN drug delivery technology capable of delivering diphtheria toxoid (DT) and tetanus toxoid (TT) in vivo [97]. The results showed that DT and TT can be successfully loaded into the tips of ceramic nanoporous MNs. By labeling the antigens with near-infrared fluorescence, they applied DT and TT-loaded nanoporous MNAs to mice skin in vivo and induced antigen-specific antibodies similar to those obtained with subcutaneous immunization at the same doses, opening the possibilities for future MN vaccination designs. Leone et al. achieved single and multiple injections of DT using dMNs and hMNs [72]. The prepared dMN can penetrate the skin and be dissolved within 20 min to release the antigen all at once. Skin immunization with hMN was performed by repeated dose injections. The overall response to dissolved MN vaccination was higher than that of hMN, and the immune strength was also high in the absence of adjuvant. In conclusion, unadjuvanted dissolving MNs were proved to be promising delivery vehicles for vaccination. Pattarabhiran et al. investigated potent immune responses elicited by tetanus toxoid-antigen-dissolvable microneedles (TT-MN) in a mouse model [71]. They prepared TT-MN by adding TT to the polymer mixture. The results showed that the MNs penetrated 130 microns deep into the mouse skin and dissolved completely within 1 h of insertion into the skin. The TT-MN group had higher antibody titers than the intramuscular injection group. This indicates that TT-MN can be developed as a minimally invasive system for percutaneous delivery of TT antigen.

## 4. COVID-19 Vaccines and Their MNs Delivery

Vaccine candidates against SARS-CoV-2 were rapidly being developed one year after the COVID-19 outbreak, and more than 2 billion people worldwide have now been fully vaccinated.

### 4.1. SARS-CoV-2

SARS-CoV-2 is a coronavirus strain of the respiratory disease that caused the COVID-19 pandemic [98]. It is a positive single-stranded RNA virus with an envelope [99] and belongs to the family Coronaviridae Betacoronavirus genus severe acute respiratory syndrome related coronavirus species [100]. SARS-CoV-2 is the seventh-known coronavirus that can infect humans. Available evidence suggests that it is a zoonotic coronavirus, with close genetic similarity to bat coronaviruses [101]. Research is ongoing as to whether it came directly from bats or indirectly from any intermediate host [102]. The virus showed little genetic diversity, suggesting that a spillover event introducing SARS-CoV-2 into humans may have occurred in late 2019 [103]. Epidemiological studies estimated that the primary reproduction number (R0) of SARS-CoV-2 was on average 2.4 to 3.4 between December 2019 and September 2020 [104]. However, some subsequent variants became more contagious, such as Delta (B.1.617.2 and AY lineage) and Omicron (B.1.1.529, BA.1, BA.1.1, BA.2, BA.3, BA.4 and BA.5 lineages).

The virus can invade the human body through the upper respiratory tract, and it uses the Angiotensin-converting enzyme 2 (ACE2) receptor to enter host cells. The main infected organs include the lungs, heart, kidneys and other major organs [105]. Human patients infected with the virus have no unique clinical symptoms, most of which are low-grade fever, weakness, oral and nasal symptoms, and dry cough, and some are accompanied by gastrointestinal discomfort [106] (Figure 6).

### 4.2. Types of COVID-19 Vaccines

In the aftermath of the outbreak of the pandemic, various scientific research units and vaccine companies have been working on developing various COVID-19 vaccines, all of which can teach the human immune system to safely recognize and block the virus. As of Oct 2022, there are more than 700 vaccines under development in preclinical or clinical trials, and more than 220 COVID-19 vaccine candidates are in development. Among them, at least 85 vaccine candidates are in human trials, and 40 vaccines have been approved by the FDA for production use (Figure 7).

There are two broad categories of COVID-19 vaccines, including whole viral vaccines and component viral vaccines similar to those described in Section 3. Whole-virus vaccines include inactivated vaccines and live-attenuated vaccines; component vaccines include recombinant subunit vaccines, viral vector vaccines, and nucleic acid vaccines (Figure 8).

### 4.3. MN Delivery of COVID-19 Vaccine

In 2020, Kim successfully delivered the COVID-19 recombinant protein vaccine using dissolving MNs, which kicked off this field [30]. As of October 2022, a total of 44 related papers have been published, as shown in Figure 9. Table 3 shows the development of different MN delivery of COVID-19 vaccine with their benefits and drawbacks.

#### 4.3.1. Whole-Virus Vaccine

Whole-virus vaccines are made by using chemical or physical methods such as formaldehyde or heat to kill the viral pathogen completely or partially, followed by purification and the addition of adjuvants. Such systems are very mature and are a massive production. It is a priority development technology for vaccines in response to COVID-19 outbreaks. Many whole-virus vaccines are currently available, such as BBIBP-CorV developed by the Beijing Institute of Biological Products (BBPI) of China Pharmaceutical Group [107]. CoronaVac (also known as the Sinovac COVID-19 Vaccine) was developed by Sinovac, a biopharmaceutical company in mainland China [108,109]. The Indian COVID-19 vaccine Covaxin (BBV152) was created by Bharat Biotechnology and the Indian Council of Medical Research [90]. The WHO included the above three vaccines on the “Emergency use list” in May, June, and November 2021, respectively.

Li et al. developed a smart mushroom-inspired printable and mildly detachable (MILD) MN platform for the efficient and convenient delivery of multiple-dose COVID-19 vaccines and decentralized vaccine information storage. The MILD system induced high levels of antibodies in vivo after loading with an inactivated SARS-CoV-2 virus vaccine, which is a promising vehicle that has the potential to help contain the COVID-19 pandemic [16].

#### 4.3.2. Recombinant Subunit Vaccine

Recombinant subunit vaccines contain only selected parts of pathogens and are very safe and stable. The expression vector containing the target antigen gene is transfected into an engineered cell line, and large quantities of the expression-target protein is purified; finally, the recombinant subunit vaccine can be prepared after adding an adjuvant. The preparation technology of recombinant subunit vaccines is mature, and there are currently more than 50 protein subunit vaccines in development [110]. In clinical trials, the overall protection rate of subunit vaccines is higher than that of inactivated vaccines, which are also suitable for immunocompromised individuals. However, their manufacturing is more complex and requires adjuvant co-injection.

The first vaccine to be successfully delivered using MNs was a recombinant protein vaccine [30]. Afterwards, Kuwentrai et al. prepared MNs with low molecular weight hyaluronic acid (HA) as support material for the delivery of S-RBD protein vaccines. HA is a naturally occurring skin substance that can be rapidly dissolved in skin tissue fluid. The results showed that the MN-based minimally invasive intradermal vaccine effectively penetrated the skin of mice, eliciting significant B-cell antibody responses and inducing interferon-gamma (IFN-γ)-based T-cell responses, which may control the rapid COVID-19 outbreaks [111]. McMillan et al. used a high-density microarray patch (HD-MAP) to deliver a SARS-CoV-2 spike subunit vaccine directly to the skin, which indicated that the vaccine was thermostable on the patch and enhanced cellular and antibody immune responses. In an ACE2 transgenic mouse model, a single dose of HD-MAP-delivered spikes provided complete protection against a lethal viral challenge. HD-MAP-delivered vaccines are superior to traditional needle and syringe vaccinations and could be an important addition to the ongoing COVID-19 pandemic [81].

#### 4.3.3. Viral Vector Vaccines

Viral vector vaccines are produced by constructing a viral vector containing the target antigen gene and then delivering the genetic material that encodes another infectious pathogen-targeting antigen to the recipient’s host cell. It provides the genetic material to express antigens in cells and can induce a powerful cytotoxic T cell response. Adenoviral vectors are the most used viral vectors for this vaccine.

This type of vaccine mainly includes AstraZeneca, Satellite V, Johnson & Johnson, Convidecia, etc. Oxford–AstraZeneca (AZD1222) is an improved non-replicating chimpanzee adenovirus vector (ChAdOx1) vaccine developed by the University of Oxford in cooperation with AstraZeneca Pharmaceuticals [112]. This vaccine was placed on the emergency use list by the WHO on 15 February 2021 [113]. Satellite V (Спутник V) is a vaccine developed and registered by the State Research Center of Epidemiology and Microbiology in Gamaleya, Russia. Johnson & Johnson’s COVID-19 vaccine is a human adenovirus-based viral vector vaccine developed by Janssen Vaccines in Leiden, the Netherlands and Janssen Pharmaceutica. Ad5-nCoV (Convidecia) is a single-dose recombinant adenovirus type 5 vector vaccine developed by CanSino Biological Co., Ltd. and the Institute of Bioengineering of the Academy of Military Medical Sciences of the Chinese People’s Liberation Army Academy of Sciences [114]. Ad5-nCoV is the only currently approved COVID-19 vaccine that can use a single-dose vaccination program and is approved in China [115]. Flynn et al. identified a dissolvable MNP for skin immunization to deliver the malaria vaccine AdHu5-PfRH5. Studies have shown that MNs can deliver low-dose adenovirus vaccines that are highly immunogenic. Moreover, the MNs also can stabilize the adenovirus vaccine [116].

#### 4.3.4. Nucleic Acid Vaccines

Nucleic acid vaccines introduce specific antigen-encoding DNA or mRNA sequences into the cells of an organism to induce an immune response, preventing and treating diseases. Compared with traditional vaccines, genetic vaccines have the advantages of convenient design, high speed, and low production cost. Currently, there are more than 30 types of COVID-19 mRNA vaccines in preclinical or clinical trials around the world.

The Pfizer–BioNTech vaccine (BNT162b2, trade names: Comirnaty, Fubitai) is an mRNA vaccine jointly developed by BioNTech in Germany and Pfizer in the United States. It is the first nucleic acid vaccine approved by the WHO for the prevention of COVID-19 [68]. The Moderna vaccine (mRNA-1273, trade name: Spikevax) was jointly developed by the National Institute of Allergy and Infectious Diseases at the Biomedical Advanced Research and Development Authority and Moderna Corporation [117]. On 18 June 2022, the U.S. Food and Drug Administration (FDA) issued an emergency authorization for the use of Moderna and Pfizer–BioNTech vaccines in infants and young children over 6 months of age [118]. The above two vaccines have launched mass vaccinations around the world; both of them have proven to be highly effective.

Caudill et al. used three-dimensional (3D) printing technology to design and fabricate multi-faceted coated (ovalbumin and CpG) MNAs. Compared with subcutaneous injection, these MNs not only enhanced vaccine retention in the skin, but also increased immune cell activation in the lymph nodes, which induced robust humoral and T-cell immune responses. CLIP 3D-printed MNs coated with vaccine components provide a useful platform for non-invasive, self-applied vaccination [119]. Yin et al. developed a detachable MNP for delivering polymer-encapsulated spike (or nucleocapsid) proteins that encode DNA vaccines and immune adjuvants. Through the intradermal delivery of a nanovaccine, stronger and longer-lasting adaptive immunity compared to traditional injection vaccines were achieved. Additionally, the detachable MNP can be stored at room temperature for at least 30 days without reducing the immune response [120]. Xia et al. then used an ultra-low-cost (<$1) handheld (<50 g) electroporation MN electrode array (“ePatch”) system for the DNA vaccination of SARS-CoV-2. The system induced robust adaptive immune responses in mice from at least 10-fold lower doses than traditional intramuscular or intradermal DNA vaccines [20]. In the subsequent study, Kapnick et al. discussed the fact that MNAs could reduce barriers to vaccine access in under-resourced settings and could facilitate the rapid deployment of vaccines [121].

**Table 3 pharmaceutics-15-01349-t003:** Temporal development of different microneedle delivery of COVID-19 vaccine with their benefits and drawbacks.

Time	Vaccine Type	MNs Type	Advantages	Drawbacks	Ref.
18 March 2020	Recombinant protein	dMN	1: Produced higher levels of neutralizing antibodies. 2: Reduced vaccine doses required to substantially reduce costs. 3: Polymer matrix helps stabilize vaccine for at least 1 month. 4: Potential for self-management.	1: Uncertainty in neutralizing antibody test results: only four weeks (reliable test is after six weeks). 2: Immunogenicity also needs to be evaluated in clinical trials.	[30]
12 October 2020	Recombinant protein	dMN	1: minimally invasive. 2: Significant antibody responses can be maintained for up to 97 days. 3: Will not cause any damage to the environment. 4: Facilitates rapid vaccination.	1: This platform is not suitable for delivering mRNA. 2: The titers of specific antibodies produced by the MN method vary widely. 3: Sterility is difficult to guarantee.	[111]
27 February 2021	Viral vector vaccine	dMN	1: Incorporation of vaccine into patch significantly improves its thermal stability. 2: MN delivery of vaccines enables. low-dose priming of immune response. 3: Allow storage and distribution without cold chain.	1: Clinical data are difficult to obtain and lack robust statistical analysis.	[116]
17 August 2021	Nucleic acid vaccines	cMN	1: Non-intrusive and self-applying. 2: Dose savings achieved. 3: Realized the combination of rapid 3D printing technology and MN vaccine formulation, providing a versatile platform to improve global immunization and healthcare.	1: Potential safety hazards caused by needle tip breakage. 2: Coated vaccine doses are limited.	[119]
26 August 2021	DNA vaccine	dMN	1: Enhanced thermal stability of the vaccine and ease of handling. 2: Did not have any noticeable side effects during in vivo vaccination.	1: Stability needs further investigation because DNA vaccine itself is relatively stable. 2: Nanovaccine needs more clinical research.	[120]
13 September 2021	DNA vaccine	sMN	1: Ultra-low-cost (<$1), handheld (<50 g), battery-free electroporation microneedle vaccination system. 2: Can save at least 10 times the vaccine dose. 3: Vaccination was well tolerated with mild, transient skin effects.	1: The electric field generated between the electrodes may cause skin and nerve irritation. 2: The immune response is only characterized by pseudoviruses, and more detailed immunological characterization is required.	[20]
29 October 2021	recombinant protein	dMN	1: Ease of self-administration, reduced cold chain dependency, and improved thermal stability. 2: Facilitates improved vaccine transportation and delivery to patients in low- and middle-income countries 3: Protection with just a single dose of the vaccine.	1: Antibody and/or cellular immunity induced by the vaccination regime needs further validation and may not be sufficient to suppress viral replication.	[81]
12 January 2022	mRNA Vaccine	dMN	1: Helping lower barriers to vaccine access in resource-poor settings 2: For contributions to the development of biomaterials for vaccine applications.	1: mRNA vaccines are prone to degradation 2: Antibody titers of MN vaccines vary, and precise dosing is challenging.	[121]
15 April 2022	inactivated virus	dMN	1: Helping vaccinators accurately record vaccination data. 2: Easily self-administered by individuals. 3: Patch base can be easily removed from the skin surface.	1: There are currently no large-scale trials of vaccine delivery in large animals or humans.	[16]

The study by DeSimone and his team found that their MNP produced 50 times more antibody titers than conventional injections [119]. Furthermore, researchers in Queensland have also shown that their patch can provide a higher immune response than traditional syringes. They mentioned that even after COVID-19 is over, having this MN technology means that the world will be better prepared for the next epidemic [81].

### 4.4. Advantages and Challenges of MN Delivery of COVID-19 Vaccine

The development of MN undoubtedly provides a convenient alternative vaccination method for addressing the COVID-19 pandemic, which is conducive to increasing the popularization rate of the vaccine, especially at this stage when COVID-19 vaccination has been vigorously strengthened. There are many advantages to using MNs to inoculate the COVID-19 vaccine. It could not only lower the inoculation dose of the vaccination, but it also induces higher antibody levels. Because it is less invasive, easy to operate by itself, and has enhanced thermal stability, it has the potential to expand vaccination rates in low- and middle-income countries and special populations (infants, the elderly, and pillow phobias, etc.), which are expected to achieve full vaccine coverage.

However, there are currently many challenges for the MN delivery of the COVID-19 vaccine. During the drying, curing, and loading process of the MNs, the loading of the vaccine is difficult to make uniform, and there is great instability. Therefore, the corresponding antibody titers are also unstable. For sMNs or hMNs, potential safety hazards may be caused by needle tip breakage. Whether in the process of manufacturing or in storage, sterility is difficult to guarantee. Most importantly, there is little clinical data to support the immune response after vaccination, and there is long way to go before practical application.

## 5. Challenges and Prospects of MNs Delivery of Vaccines

MN vaccination solves the cold chain storage problem of vaccines to a large extent, saves vaccine dosage, and increases the accessibility of vaccination. Research on MN vaccination has become more and more extensive in recent years, and it is expected to become a new trend for future vaccination. However, there are still many challenges for the clinical translation and large-scale production of MN vaccines.

### 5.1. Vaccine Waste

Multiple reports have demonstrated that MN delivery of smaller amounts of vaccine can achieve similar immune responses to high-dose intramuscular injections [28,119]. However, there is always a certain amount of vaccine loss in the drying and molding process of MNs, especially for cMN and dMN, which need strict molding and drying conditions [46,122]. These problems may be solved by finding more suitable MN materials and vaccine formulations.

In addition, vaccines may also cause losses during delivery. For example, after the hollow microneedle is pierced into the skin, the pinhole may be pressed by the skin tissue, making it difficult for the subsequent liquid or solid vaccine to penetrate into the skin, and the vaccine may be lost when the pillow is pulled out. For such cases, measures such as prolonging the delivery time and reducing the solvent volume should be adopted.

### 5.2. Vaccine Safety

As mentioned earlier in the article, strict control conditions are required for the production and storage, regardless if it is a full-dose or a component vaccine. The use of MNs to deliver vaccines increases thermal stability to a certain extent, but the sterilization of MN vaccines is challenging [111]. The sterilization of sMN is straightforward, and commonly used methods include dry heat sterilization, moist heat sterilization, gamma radiation, etc. [123], but it is usually not suitable for delivering vaccines. Vaccines are fragile bioactive components that need to be introduced into the body, usually using cMN or dMN for delivery. During the sterilization process, not only do the stability and activity of the vaccine components need to be maintained, but the supportability of the MN material also must be ensured. There are great challenges in the selection of sterilization methods, and some chemical and physical methods are almost not applicable [124]. McCrudden et al. sterilized dMN loaded with the model drug ellipsoid protein (OVA) in different ways, including gamma irradiation, moist heat, and dry heat sterilization. It was found that the OVA components were either destroyed or not detected at all [124]. Current studies have found that phthalene and electron beam sterilization may be effective methods for MN sterilization, but they cannot achieve complete sterility [125]. Sterilization requires extensive research before entering commercial production and approval, which is one of the most important challenges in MN-delivered vaccine systems.

The immunogenicity of the skin makes it a highly sensitive organ for MN vaccine delivery, but MNs could also cause side effects such as mild and temporary erythema to the skin during the insertion process [126]. This is one of the safety issues that should be considered with all types of MN vaccines. Before clinical trials, this issue must be strictly tested and evaluated.

### 5.3. Vaccine Degradation

Dissolving MNs can generally be dried at a relatively mild temperature (25–40 °C) in the molding process, which will limit their applications for some vaccines. For example, recombinant protein vaccines are easily degraded when placed at room temperature for a period [70]. In addition, the temperature can reach 100 °C when curing MNs, especially during the methods of photolithography and wire drawing, which are likely to degrade the corresponding vaccine [127]. For some heat-sensitive antigens, the temperature is even more critical.

### 5.4. Vaccine Loading

Most microneedle vaccines have a relatively limited loading capacity, except for hMN, which can only achieve continuous infusion or “on-demand” administration. CMN can only deliver about 1 mg of vaccine, while sMN and hydMN deliver vaccines depending on their needle volume and storage capacity [69,128]. After the hMN are inserted into the skin, the central outlet may be blocked by compressed skin tissue. Despite the potential of MNs to overcome the barrier properties of the skin, their success largely depends on the passive diffusion of vaccines in the skin. Other types of MNs also rely on this to deliver vaccines. Vaccines in particular require a threshold dose to induce immunity, and when passive diffusion is relied upon, antibody responses become erratic [129]. This is also one of the insurmountable obstacles for MN vaccines. It requires continuous improvement of technology and manufacturing, the repeated testing of vaccine loads, and the use of large amounts of data to evaluate the degree of absorption of vaccines and other methods to solve the problem.

### 5.5. Prospects

Despite the aforementioned limitations and challenges of scaling up production, the commercial use of MNs to deliver vaccines has existed for more than a decade. Continuing advances in MN technology have enabled the painless delivery of a variety of vaccines. The drivers and potential benefits of MNs in vaccination are clear, including dose savings, elimination of the cold chain, improved safety, and potential self-administration. MN vaccines could greatly overcome vaccination barriers, eliminate the problem of unfair access to vaccines, and bring good news for the vaccination of special groups.

#### 5.5.1. Practical Application of MNs Vaccines

The evidence and support for MN vaccination systems have grown in recent years. In 2020, the Vaccine Innovation Priorities Strategy (VIPS) Alliance [Bill & Melinda Gates Foundation (BMGF), Garvey, UNICEF, PATH, WHO] recommended MN as one of three priority innovation to help overcome barriers to vaccination, ensure equitable access, improve vaccine effectiveness, and contribute to global health security [130]. MNs are expected to be suitable for the delivery of different types of vaccines. The U.S. government provided USD 430,000 to four groups for developing MNP through the Biomedical Advanced Research and Development Authority (BARDA) on 18 August 2020 [131]. Vaxxas is actively developing MN vaccines and is actively facing challenges with some partners, including BMGF, Merck and BARDA [132]. Vaxxas is advancing an easy-to-use device for the MN delivery of IPV vaccines, designed to deliver vaccines into the skin. They observed significant vaccine dose savings in preclinical models. They also demonstrated this benefit in a Phase 1 clinical study, where the MN delivered a dose of seasonal fluluenza vaccine that required only one-sixth the standard dose for intramuscular injections [133].

In addition, an easy-to-use vaccine-delivery MNP from the University of Connecticut (UConn) delivers the S protein on the coat of SARS-CoV-2 into the skin as an antigen against COVID-19. The technology could have far-reaching implications for improving vaccine coverage. On 17 May 2022, Emergex Vaccines Holding Limited (Emergex) announced the successful coating of its novel CD8+ T cell adaptive COVID-19 vaccine onto Zosano Pharma’s MNP. Zosano Pharmaceuticals has designed and manufactured a proprietary MNP system that consists of approximately 2000 drug-coated titanium MNs, and they successfully applied the Emergex COVID-19 vaccine candidate to MNs [134]. Emergex and Zosano have demonstrated that Emergex’s COVID-19 vaccine is stable on plaques over a broad temperature range for up to six months at 40 °C/75% relative humidity. Minimizing cold chain logistics could reduce the cost and efficiency of vaccination programs around the world and improve global preparedness for future pandemics and disease outbreaks [135].

Micron Biomedical has also begun testing a measles-rubella vaccine based on dMN in children. The researchers assessed the safety, tolerability, and immunogenicity of the measles and rubella vaccines using MN technology compared with standard subcutaneous injections. It is worth mentioning that Micron chose Gambia, a sub-Saharan African country, as the clinical trial site, and enrolled 120 infants aged 15 to 18 months and 120 infants aged 9 to 10 months. This reflects a belief that MN technology is particularly well-suited for the administration of vaccines in areas of the world with special needs, as well as for people with special needs [136].

#### 5.5.2. An Alternative Solution of Mass Vaccination in Special Population

While the world is encouraging vaccinations, vaccine coverage is still relatively limited, especially for special populations, which are also the main morbidity and mortality groups. The importance of vaccine equity has been made painfully clear by the COVID-19 pandemic; unvaccinated people are 15 times more likely to die from the disease than fully vaccinated people [137]. Vaccines have all but eliminated infections such as measles, tetanus and polio. However, lack of access to a vaccine means these diseases remain a very real threat in many countries. Mark Prausnitz, a biomolecular engineer at the Georgia Institute of Technology, describes the measles shot as “cheap”. However, 15% of children worldwide do not even receive the first dose. Tens of thousands of children under five die each year from this preventable disease, almost exclusively in low- and middle-income countries [138].

Courtney Jarrahian, who works on vaccine delivery at PATH, a global health nonprofit, said “The pandemic has shown us something very important, that diseases can spread and worsen in unvaccinated populations. Only when everyone has equal access to life-saving vaccines, health and safety around the world could be ensured. The key to ensuring that everyone has equal access to vaccines is to ensure that children in remote areas, vulnerable groups, and elderly people with limited mobility and poor physical fitness also have access to life-saving vaccines” [139].

The current FDA and European Commission (EC)-approved COVID-19 vaccines must be refrigerated or frozen, so some remote or low-income countries and regions have extremely limited vaccine resources, and fewer people there can be vaccinated. If the use of MNs could get rid of the restrictions of the cold chain and reduce the number of vaccine preparations, people in these areas will have more opportunities to achieve vaccination. Additionally, the vaccination must be administered intramuscularly by trained medical staff in a qualified medical institution; some elderly or disabled people with limited mobility cannot be vaccinated without the companionship of their family members, but they happen to be a group of people with a weakened immune system who are in desperate need of vaccination [140]. The emergence of MN vaccines could be one of the most efficient and convenient ways for them to get vaccinated. Just like sticking on a Band-Aid, they can achieve immunization by applying a little force on their arms or legs. Moreover, both fears of needles and the pain of immunizations prevent many children and people who are extremely sensitive to pain from getting vaccinated. Although some areas, such as Hong Kong, have opened the vaccination of COVID-19 to babies aged 6 months to 4 years, and a lot of evidence has proved that the vaccine is safe enough for babies, many parents still refuse to vaccinate their babies, largely because of the needle pain inflicted on the baby and the inconvenience [141]. In addition, immune cells are not dense in muscles, and multiple injections, needle stick injuries, and unsafe injection procedures can lead to the risk of serious health problems, adding to the idea of parents refusing to vaccinate their babies. The skin is rich in APCs, so scientists have been investigating the use of MNs to deliver vaccines safely and painlessly into the skin, which not only eliminates many health risks, but also promises to achieve all the effects of the desired immune response. The provision of MN vaccines is now one of the top priorities, and more research is needed to explore its benefits in detail, as mentioned in the WHO report [142].

## 6. Conclusions

Interest in MNs has grown since its first appearance in the late 1990s, and many research groups and companies have attempted to efficiently manufacture various types of MNs; thus, they have emerged as promising transdermal drug and vaccine delivery devices. Vaccines are an essential part of pandemic preparedness, but their coverage is severely limited. Comprehensive vaccine coverage is extremely critical, especially for groups in remote areas and vulnerable groups (infants, young children afraid of pain and elderly people with limited mobility). Because of the supply chain and delivery challenges, pain and panic from needle sticks, needle contamination, and a lack of trained medical professionals, the emergence of MN delivery of vaccines has brought good news. Although technology, production and supervision challenges for MN vaccines remain, researchers and companies are actively joining in on new technology exploitation to facilitate MN vaccine inoculation. They work to provide boosters and alternatives to pandemic and routine vaccination.

The MN vaccine is expected to greatly increase the global vaccination rate and help meet the demand for full vaccine coverage. Moreover, having MN vaccination technology could help the world be better prepared for the next pandemic.

## Figures and Tables

**Figure 1 pharmaceutics-15-01349-f001:**
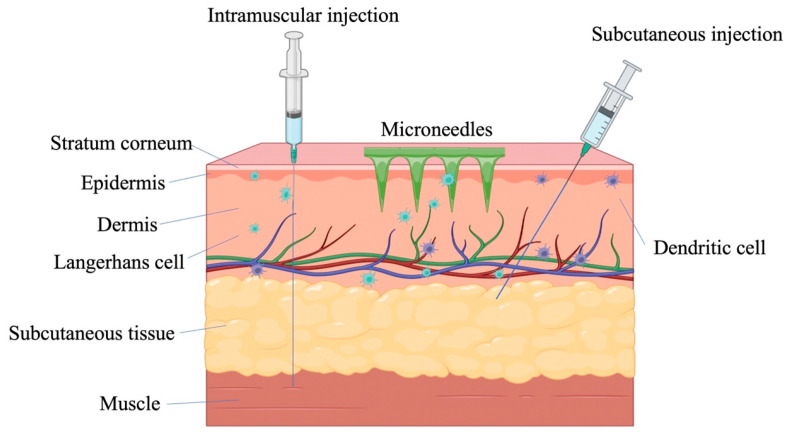
Different routes of vaccination.

**Figure 2 pharmaceutics-15-01349-f002:**
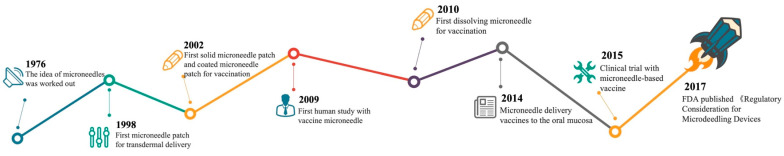
Timeline of Microneedle Research Development.

**Figure 3 pharmaceutics-15-01349-f003:**
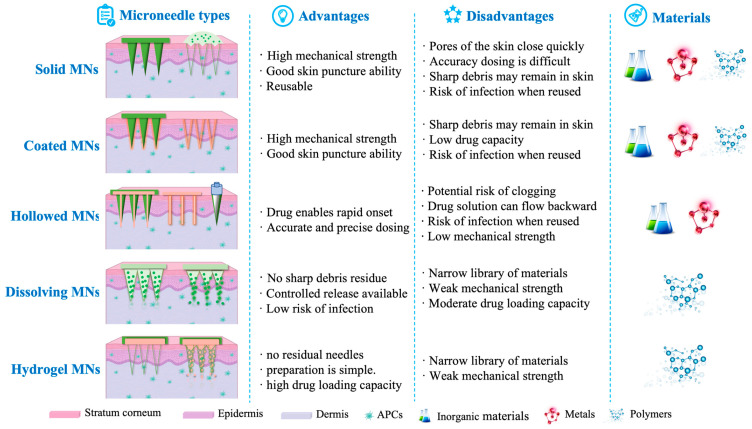
Overview of MN types.

**Figure 4 pharmaceutics-15-01349-f004:**
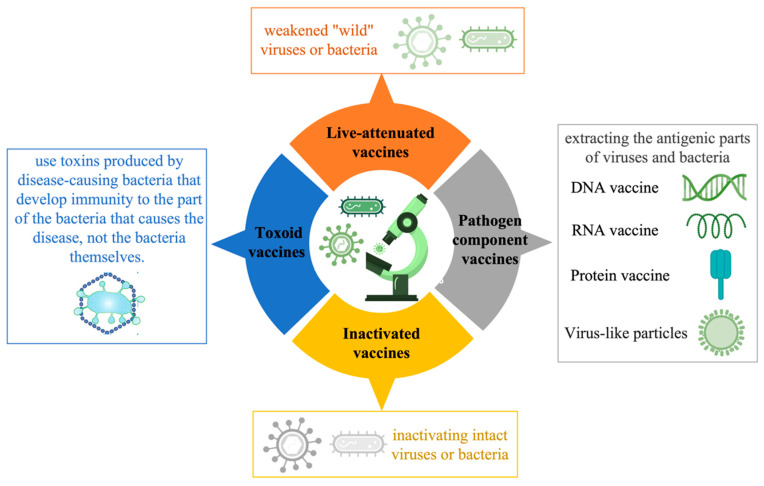
Classification of vaccines.

**Figure 5 pharmaceutics-15-01349-f005:**
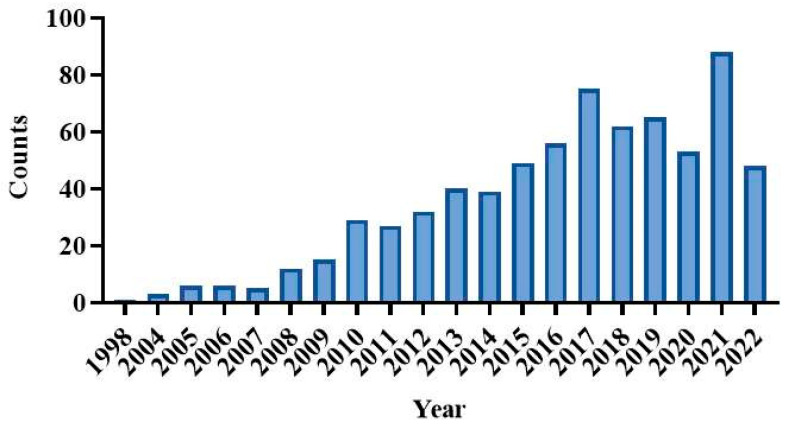
Cumulative number of publications on microneedles for vaccination. The total number of publications was determined by searching the PubMed database (http://www.ncbi.nlm.nih.gov/pubmed/) on 15 October 2022, using the search terms “microneedle for vaccine”.

**Figure 6 pharmaceutics-15-01349-f006:**
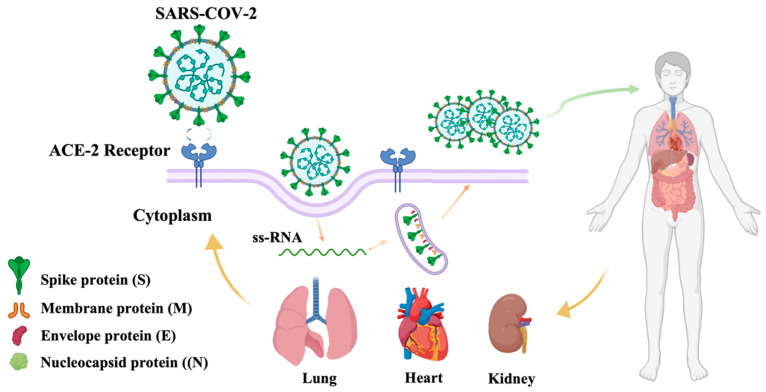
Structure and infection pathways of SARS-CoV-2.

**Figure 7 pharmaceutics-15-01349-f007:**
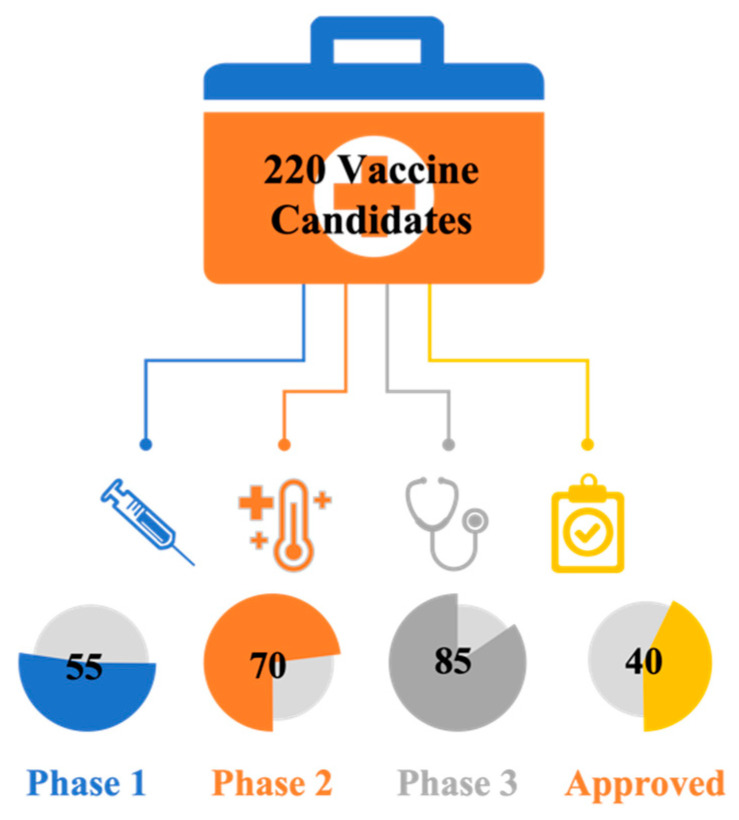
COVID-19 Vaccines Candidates in Clinical Trials.

**Figure 8 pharmaceutics-15-01349-f008:**
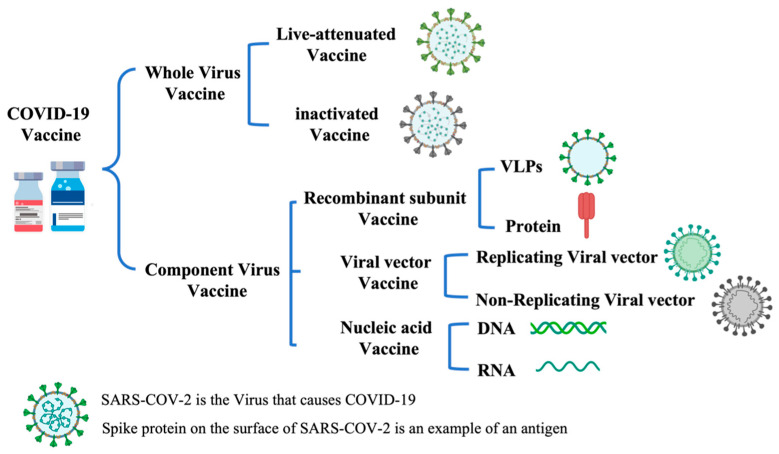
Classification of COVID-19 vaccines.

**Figure 9 pharmaceutics-15-01349-f009:**
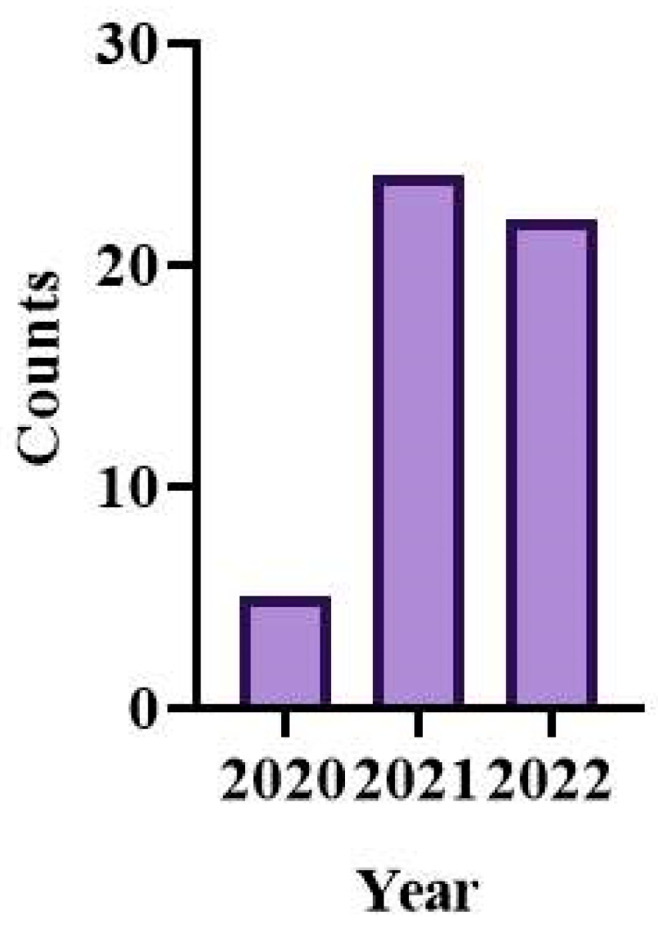
Cumulative number of publications on microneedles for COVID-19 vaccination. The total number of publications was determined by searching the PubMed database (http://www.ncbi.nlm.nih.gov/pubmed/) on 15 October 2022, using the search terms “microneedle COVID-19”.

**Table 1 pharmaceutics-15-01349-t001:** The differences between microneedle delivery of vaccine and traditional injection vaccines.

Description	Traditional Injection Vaccines		MN-Based Vaccines				
Types	Intramuscular	Intradermal	sMN	cMN	hMN	dMN	Hyd MN
Vaccines	Influenza [17], HBV [18]	BCG [19]	COVID-19 [20]	Ebola [21]	Polio [22]	HPV [23]	Ovalbumin [24]
Sharps injury and contamination	Have risks		Needle Breakage Risk			No risks	
Stability	Refrigerated or frozen		Depending to the vaccine			Room temperature	
Cold chain	Need		Depending to the vaccine			No need	
Mechanism	Vaccines are injected into the muscle or intradermal multiple times through a syringe		Vaccines bypassing the stratum corneum and directly into epidermis or dermis				
Pain	Yes		Pain-free				
Patient compliance	Non-compliant		compliant				
Self-administration	Inoperable		Operable				

**Table 2 pharmaceutics-15-01349-t002:** Typical examples for each type of vaccine.

Type	Vaccine	Virus	Reference
Live-attenuated	Poliovirus vaccine	Poliomyelitis	[60]
	Chimeric Flavivirus vaccine	Japanese encephalitis	[61]
	Measles vaccine	Measles virus	[27]
	Rubella vaccine	Rubella virus	[62]
	BCG	Mycobacterium tuberculosis	[14]
Inactive vaccine	Hepatitis A	Hepatovirus A	[61]
	Hepatitis B	Hepatitis B virus	[63]
	Influenza vaccines	Influenza virus	[64]
	Pseudomonas aeruginosa vaccine	Pseudomonas aeruginosa	[65]
DNA vaccine	Rabies DNA vaccine	Rabies	[28]
	Cancer vaccine	Malignant melanoma	[66]
	Hepatitis B	Hepatitis B virus	[67]
RNA vaccine	COVID-19 vaccine	Sars-Cov-2	[68]
Protein vaccine	F1 protein antigen of Yersinia pestis vaccine	Plague	[69]
	Recombinant subunit Influenza vaccine	Influenza virus	[70]
VLP vaccine	Influenza VLPs vaccine	Influenza virus	[26]
	HPV vaccine	Human papillomavirus	[23]
toxoid vaccines	Tetanus vaccine	Tetanus	[71]
	Diphtheria vaccine	Corynebacterium diphtheriae	[72]

Measles: Measles is an acute and highly contagious viral disease caused by infection with measles virus. It is common in children and is one of the common acute respiratory infectious diseases in children. Rubella: Rubella is an infection caused by the rubella virus, the main symptom is a rash on the face which spreads to the trunk and extremities. Hepatitis A: Hepatitis A is an infectious disease of the liver caused by Hepatovirus A (HAV); it is a type of viral hepatitis. Hepatitis B: Hepatitis B is an infectious disease caused by the Hepatitis B virus (HBV) that affects the liver; it is a type of viral hepatitis. It can cause both acute and chronic infection.

## Data Availability

The data presented in this study are available on request from the corresponding author.

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
