# Peer review of "Microneedles: An Emerging Vaccine Delivery Tool and a Prospective Solution to the Challenges of SARS-CoV-2 Mass Vaccination"

_pharmaceutics, 2023, doi:10.3390/pharmaceutics15051349_

Round 1
Reviewer 1 Report
pharmaceutics-2347342-peer-review-v1
An interesting review paper that deserves to be more specific, the authors see great potential for the Microneedles, and the reader can share this enthusiasm, but the article sometimes turns into a panegyric, I mainly recommend correcting this and discussing further the disadvantages related to MNs, in particular for a pandemic. Many generic (and thus wrong) assertions are made (metals are biocompatible…), I let the authors double-check this using the detailed list of comments/recommendations provided hereafter:
Introduction: it could be great to add some information about current vaccine practices using the intradermal route (BCG), the Mantoux technique…
lines 65-66: “is able to stimulate”: the effect is not systematic, I suggest being more careful here and mentioning a specific clinical study
Line 67: same comment: not all mNeedles store vaccines in an anhydrous form, please reformulate.
Table 1 presents the differences between mNeedles and traditional injections in a way that is not very fair. This is a very idealized and biased view of injection with mNeedles, please add some contrast to this black-and-white view of the problem, also the mNeedle characteristics strongly depend on the MN types, and Table 1 should reflect this (Fig 3).
Line 82: typo; also, please indicate if the results were obtained with animals or humans (this comment applies to the whole section).
Line 107: single mNeedle is also available for vaccination (see system made by Debiotech SA)
Lines 116 to 121: a very short introduction to MNs’ history, could be improved.
Line 154: this sentence is in contradiction with Table 1: please revise.
Figure 3: hollow MNs can be connected to a syringe that generates pressure to inject the dose, thus the pb of backflow is not systematic.
Line 198: “silicon is bioincompatible” is it true? Further silicon surface always shows silicon dioxide and not pure silicon, please revise.
Line 199: “production is expensive”: this is correct for small series, MEMS is cost-effective at large volumes, please revise.
Line 207: not all metals are biocompatible, far from it! Please revise.
Same comment for lines 214-215.
Line 225: Please moderate a bit your enthusiasm for MNs!
Section 3.1: a table with typical examples for each type of vaccine may help (influenza, BCG, Measles, Covid…)
Line 236: incomplete sentence.
Line 397: “lower dose”: which percentage?
Line 496: do you mean: compatible with mass production?
Line 559: please remove the duplicate.
Section 5: the authors focus on coated and dissolving MNs but other types of MNs are possible (hollow). I suggest the authors double-check if some trials have been made using hollow MNs (Covid), and briefly discuss the advantage of using them (cold chain will remain an issue but there is no need for reformulation, a process that can take years to be validated).
I also suggest the authors elaborate more about this section which is a bit short compared to the core of the paper. Economical aspects and industrialization are key factors that should be better discussed. In particular, huge production factories were built during the pandemic, It is not clear that manufacturers (and governments) will adopt alternative and risky options, as they need to amortize their investments.
A discussion about the delivery timeframe and other practical concerns could help (time spent per patient using standard vaccine process or MNs…).
Instead of using broad considerations (e.g. line 616: “the recovered antigen cannot guarantee the quality”), I suggest being more specific and using data from the literature listed in Table 3).
Line 681: ref?
Line 693 (last sentence): good to see the use of conditional mode! Again, this mode should be widely employed in the whole paper as all studies are still very preliminary and the time to market is unknown.
During revision, please take care to check that the new conclusion reflects the modification made in the core of the text.
See the previous section Comments for authors
Reviewer 2 Report
The authors have written a very interesting review article. The manuscript is well written and easy to understand, however a few grammatical errors and typos are found in it. Please see comments below for improving the quality of this manuscript.
- Abstract:
(1) Did the author discuss about the kinetics? Please replace this term with the more suitable one.
(2) Please provide the full term before using the abbreviation.
(3) The keywords, have not been written alphabetically.
- Introduction & manuscript body:
(1) The authors should also explain that muscle is not a good target for vaccination since the presence of the immune system is not as much as found in the dermis.
(2) It is written in Page 2:"MNs store vaccines in an anhydrous form" Please clarify this statement, which type of MN is this? Different type of MN uses different state of vaccines.
(3) Table 1: Do vaccines target the cells in the epidermis?
(4) Page 3:"Several important milestones in MNs development are illustrated." The authors should mention what type of MN used in those studies.
(5) Figure 3: "Moderate drug loading capacity" in hydrogel MN. Since hydrogel MNs uses drug containing reservoir, this statement is ambiguous. Please justify this.
(6) Some references are required, for instance in Page 6 paragraph 2.
(7) Figure 5: The figure should be revised for a better readability. It is quite difficult to read the graph, because the x-axis and the bar graph does not match.
(8) MN Deliveryof COVID-19 vaccine: The authors should explain the benefits and drawbacks, or even possibility of using different MN types for delivering Sars-cov vaccines.
(9) The challenges should be completed with the industrial scale production of the MN vaccines. What are the issues? How to overcome these issues.
(10) The authors should write more discussion regarding the potential and risk of MN vaccine development.
(11) A general conclusion is also required.
The English is quite good with minor corrections of grammatical errors.
Reviewer 3 Report
The manuscript by Feng Yaxiu reviewed the current progress of microneedle-based vaccine delivery tool to fight against COVID-19. The author first systematically reviews the microneedle technology, including its development, classification, materials used, and also the development of microneedle-based delivery for different types of vaccines, such as live-attenuated vaccine, inactive vaccine, and DNA/RNA/Protein/VLP/toxoid vaccine. A brief review on different types of COVID-19 vaccine is also included in the manuscript. And finally, the author compiled the current progress of development of different types of microneedle delivery for COVID-19 vaccine. The author also summarizes the challenges and prospects of microneedle-based vaccine delivery. Overall, the manuscript is well written and organized, contains a high level of in-depth summarization and analyses, and provides an important contribution towards microneedle-based vaccination technology, particularly for COVID-19 vaccine. Thus, I would like to recommend the manuscript for publication.
Author Response
Thank you for your kind comments and affirmation. In this review, the ongoing development and application of MNs based vaccines are presented. We have focused on the lessons learned from the pandemic, hoping to popularize vaccination through MNs technology and bring certain protection to social health.
Round 2
Reviewer 1 Report
I would like to thank the authors for conscientiously considering all the comments raised during the first revision of the manuscript. The revised version of the manuscript may be published in Pharmaceutics.
Reviewer 2 Report
The manuscript has been well revised and clearly explained.